# Cystic Neutrophilic Granulomatous Mastitis Treatment with Consecutive Dapsone and Adalimumab

Samir Kamat [1] , William Schaffenburg [2] and Michelle Bongiorno [2,*]

1   Department of Dermatology, Icahn School of Medicine at Mount Sinai, New York, NY 10029, USA
2   Department of Dermatology, Walter Reed National Military Medical Center, Bethesda, MD 20814, USA
*   Correspondence: dr.michellebongiorno@gmail.com

**Abstract:** Cystic neutrophilic granulomatous mastitis is a rarely reported, benign, mastitis that is not associated with lactation. The pathophysiology is still unknown; however, there is often a relationship with *Corynebacterium kroppenstedtii*. Cystic neutrophilic mastitis can have deep seated tender nodules and cutaneous ulceration which can clinically resemble pyoderma gangrenous. It can be treatment refractory and can progress to a point where mastectomy is required. This case series reports two patients treated with adalimumab with remission of disease. One patient first received dapsone with remission of symptoms, but incompatible side effects resulted in discontinuation. Both dapsone and adalimumab appear to provide disease remission in patients with cystic neutrophilic granulomatous mastitis.

**Keywords:** mastitis; breast ulcerations; immunosuppression; breast infection





## 1. Introduction

Mastitis is frequently thought of as a polymicrobial infection seen in breastfeeding women, often treated with antibiotics or surgical incision and drainage. However, there are subtypes of mastitis which affect non-lactating women that are descriptively named based on their histological findings. One subtype of mastitis has granulomatous inflammation and can be due to multiple causes including sarcoidosis, reaction to an exogenous material, and infection [1]. When these causes have been ruled out, there is the encompassing term "idiopathic granulomatous lobular mastitis (IGM)".

The definitive pathophysiology of IGM remains unclear, but there are several inflammatory and infectious associations, including trauma, amyloidosis, hyperprolactinemia, granulomatosis with polyangiitis, and fungal infection [2]. IGM typically affects reproductive-aged women, often within the years following parturition, with a typical presentation of unilateral, firm, tender breast plaques that can mimic inflammatory breast cancer [2]. A subtype of IGM has been coined as "Cystic neutrophilic granulomatous mastitis" (CNGM), which commonly manifests as a breast mass with nipple inversion or sinus formation, with symptoms including: pain, nipple discharge, erythema, and abscess formation [3].

CNGM is further characterized (and differentiated by some sources from IGM) by suppurative lipogranulomas, which consist of acute to chronic granulation tissue, with central lipid vacuoles surrounded by neutrophils. Additional inflammatory cells can also be present, including eosinophils, histiocytes, and multinucleated giant cells. Necrosis may be focally present but is not a defining feature. Within these spaces, lipophilic Gram-positive *Corynebacterium* (e.g., *C. kroppenstedtii*) species are commonly reported [4].

The first-line treatment for CNGM is commonly antimicrobial medications [5]. Most women will also receive immunosuppressants (e.g., prednisone) alongside an antibiotic. In some cases, sustained use of these medications can result in disease remission. However, cases that are refractory to medical treatment may have to proceed to surgical resection of breast tissue.

Past success in managing granulomatous mastitis with steroids suggests that IGM has an autoimmune component [3]. One promising immunosuppressive, in particular, is Adalimumab, a tumor necrosis factor-$\alpha$ inhibitor (TNFa-i). Additionally, dapsone may provide benefits by it antimicrobial and anti-inflammatory properties. Thus, following conventional therapy typical for CNGM, we treated one patient with consecutive treatments of dapsone and adalimumab with remission of symptoms. A second patient was treated with adalimumab alone with remission of symptoms.

## 2. Case Report

### 2.1. Patient 1

A 38-year-old female with Fitzpatrick type III skin presented to the dermatology clinic with a tender, warm, erythematous 5 cm plaque on the left breast. She was evaluated for both infectious and malignant processes with fine-needle aspiration, two-punch biopsies, and tissue culture. Histologically, her biopsies demonstrated numerous focally necrotic granulomata with negative fungal, acid-fast bacterial, and Gram stains (Figure 1) At that time, a diagnosis of CNGM was made. Treatment initially consisted of doxycycline 100 mg twice daily and oral prednisone. Despite prolonged courses of prednisone, her disease was not controlled with tapering to doses lower than 40 mg daily.

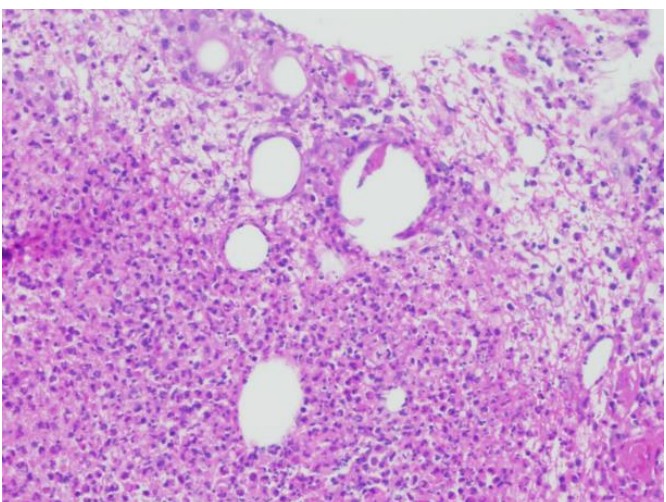

**Figure 1.** Hematoxylin and eosin staining at 20× magnification, showing cystic spaces with granulomatous inflammation and mixed neutrophilic and lymphohistiocytic inflammation.

Nine months after diagnosis, the patient was bridged with prednisone onto adalimumab. She was given a loading dose of 80 mg on day 0, followed by 40 mg weekly. This provided remission of symptoms, and the patient noted complete resolution of her disease in the left breast. She continued with the regimen for 18 months with well-controlled disease but noted prominent fatigue while on adalimumab which could not be attributed to a separate etiology. Due to fatigue, she self-discontinued the medication, which resulted in recurrence of symptoms in the left breast. On repeat analysis, it was again found to be consistent with granulomatous mastitis. She subsequently declined any further medical management.

### 2.2. Patient 2

A 40-year-old Fitzpatrick type II female presented to dermatology clinic with a tender, warm, erythematous nodules on the right breast (Figure 2) Multiple fine-needle aspirations and biopsies were obtained demonstrating acute and chronic inflammation, with sections showing mixed inflammation, including neutrophils, eosinophils, and granulomata with admixed multinucleated giant cells and associated collections of Gram-positive bacilli, most consistent with *Corynebacterium* species. Over the course of the next two years, she

was treated with intermittent bursts of prednisone and continual antimicrobial therapy with doxycycline or clarithromycin. During this time, she had several samples obtained for both histologic and microbial evaluation with similar findings to prior biopsies, with varying levels of Corynebacteria present, depending on systemic antibiotic regimen. Initial treatment with prednisone and doxycycline was tolerated with fair control, and was subsequently changed to prednisone and clarithromycin with similar results. Due to desired improvement in symptoms, a trial of colchicine was attempted with resultant worsening in disease control. She was subsequently placed on dapsone for 20 weeks with overall improved disease control. However, as a result of prominent sensory and motor neuropathies, she was subsequently transitioned onto adalimumab with dosing similar to our previous patient, that is, a loading dose of 80 mg on day 0 followed by 40 mg weekly. This has provided disease remission to date, which is currently 20 months.

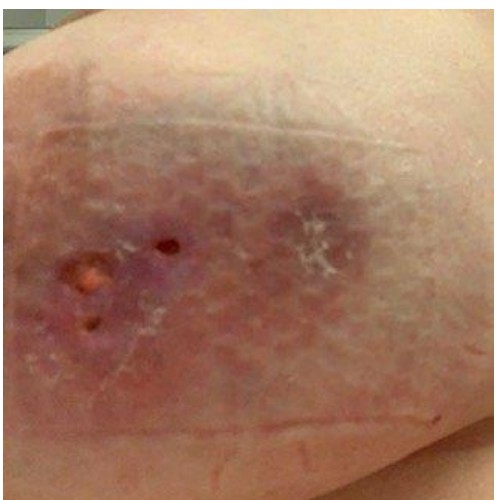

**Figure 2.** Left lateral breast with 5 cm oval erythematous indurated plaque with firm palpable subcutaneous tissue. Three overlying 2–5 mm round ulcerations, one with adherent yellow fibrinous debris at the base.

## 3. Discussion

CNGM typically occurs in parous women during the post-lactational period and is more prevalent among Māori, Pacific Islanders, Hispanic, Chilean, and Southeast Asian populations [6]. While CNGM and IGM share an identical clinical presentation, they differ in histological presentation [7]. Therefore, clinical and histologic evaluation are required to confirm the CNGM diagnosis. However, the low occurrence of CNGM and its resemblance to many other conditions (e.g., invasive carcinoma) makes it an often missed diagnosis [3].

CNGM consists of lipogranulomas that have central lipid vacuoles surrounded by neutrophils, and an outer cuff of epithelioid histiocytes [3]. Of note, the surrounding infiltrates consists of Langhans-type giant cells, lymphocytes, and neutrophils [3]. These lipid vacuoles may sometimes have rod-shaped Gram-positive bacilli [3]. These bacteria, also termed "diphtheroids" are aerobic (and facultative anaerobic), catalase-positive, asporogenous, nonmotile, club-shaped bacilli that lack the mycolic acid production of other *Corynebacterium* species [5]. One study found that performing 6 μm thick section Gram stains of representative tissue blocks enhances detection rates and facilitates identifying Gram-positive bacteria in CNGM [8].

The typical management approaches have drawn on surgery, antibiotics, steroids, or immunosuppressives [9]. Unfortunately, there is not a mainstay treatment that has achieved full symptom remission or cure. Because of the lipophilic nature of these cystic spaces, water-soluble β-lactams are not effective in targeting these bacteria [8]. Conversely, lipophilic antimicrobials have shown promise in inducing remission of symptoms, specifically including clarithromycin, trimethoprim-sulfamethoxazole, penicillin, doxycycline,

erythromycin, clindamycin, rifampin, and tetracycline [5]. It is recommended that a course of antimicrobial therapy be initiated once the diagnosis of CNGM is made, regardless of bacterial status on histology and from tissue culture.

Past successes with the use of steroids and association with autoimmune conditions suggest IGM has an autoimmune cause [3]. For example, for IGM treated with methotrexate, there is a 57–100% rate of remission-free relapse, and one study included two patients whose pathological presentation was consistent with CNGM [10]. While biologics and immunosuppressants are not typical of a breast surgeon's approach, they are regularly used by dermatologists. In fact, biologics have not been extensively incorporated in the treatment of CNGM, though they have been regularly used for histiocytic and neutrophilic dermatoses.

Tumor necrosis factor-alpha (TNF-$\alpha$) inhibitors are an established long-term treatment for chronic granulomatous autoimmune diseases, including sarcoidosis and inflammatory bowel disease [9]. One study identified higher levels of proinflammatory cytokines (IL-8 and IL-17) in cases of IGM compared to controls [9]. Although the levels of TNF-$\alpha$ have not been found to be elevated in cases of INGM/CNGM, other T helper 17/IL-17 diseases (e.g., psoriasis) and elevated IL-8 diseases (rheumatoid arthritis) have responded well to TNF-$\alpha$ inhibition [9]. Different from steroid-based, anti-microbial, or surgical approaches, TNF-$\alpha$ inhibitors are a potential alternative for managing CNGM [9]. One TNF-$\alpha$ inhibitor in particular is Adalimumab, which has been employed in the treatment of autoimmune conditions such as plaque psoriasis and other autoinflammatory diseases for decades [9].

In literature review, two prior documented cases were found to have used TNF-$\alpha$ inhibitors in the treatment of IGM/CNGM [9]. In one case report, one patient with IGM was successfully managed with dual therapy of methotrexate and etanercept in order to manage her progressive breast inflammation, which worsened when the immunosuppressants were suspended [11]. In a second case report of CNGM treated by adalimumab, treatment resulted in rapid improvement in symptoms, with evidence of disease control, as the patient's symptoms worsened with the discontinuation of medication and improvement following reintroduction [9].

Whereas adalimumab targets the histiocytic component of CNGM, dapsone targets the neutrophilic activation of the disease through anti-neutrophilic properties. Dapsone has antibiotic activity through its competitive role with para-amino benzoic acid (PABA) in the folate synthesis pathway. However, the anti-inflammatory properties remain uncharacterized. In vitro models suggest that dapsone inhibits the expression of $\beta$-2 integration (CD11/CD18) adhesion of neutrophils. This may account for dapsone's effectiveness in treating dermatitis herpetiformis and Sweet syndrome, conditions that have considerable neutrophilic infiltration. One case report found that a patient with IGM refractory to prednisone therapy was effectively managed with dapsone.

Herein, we demonstrate the effectiveness of therapy with dapsone and adalimumab, with the known side effects of dapsone limiting further use. Resolution in two patients of purulent ulceration, painful sores, and granulomata was achieved following adequate treatment with adalimumab but with unfortunate residual scar formation. Patient 1, who experienced disease recurrence upon stopping adalimumab was subsequently lost to follow up. Patient 2 experienced sensory neuropathy, a well-known side effect of dapsone, which prompted discontinuation; meanwhile, patient 1 had generalized fatigue associated with the dapsone.

## 4. Conclusions

Based on our experience, both dapsone and adalimumab appears to be a well-tolerated and safe treatment for CNGM. Limitations to this report include the number of patients, which is not generalizable. Future studies can incorporate double-blinded larger case studies and prospective design.

**Author Contributions:** Conceptualization M.B.; methodology, M.B.; validation, S.K., W.S. and M.B.; formal analysis, W.S. and M.B.; investigation, M.B.; resources, S.K., W.S. and M.B.; writing—original draft preparation, S.K., W.S. and M.B.; writing—review and editing, S.K., W.S. and M.B.; visualization, S.K., W.S. and M.B.; supervision, M.B.; project administration, M.B. All authors have read and agreed to the published version of the manuscript.

**Funding:** This research received no external funding.

**Institutional Review Board Statement:** Ethical review and approval were waived for this study due to the nature as a retrospective case series.

**Informed Consent Statement:** Written informed consent has been obtained from the patient(s) to publish this paper.

**Conflicts of Interest:** The authors declare no conflict of interest.

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
