# Peer review of "Cystic Neutrophilic Granulomatous Mastitis Treatment with Consecutive Dapsone and Adalimumab"

_dermatopathology, doi:10.3390/dermatopathology9040047_

Round 1

Reviewer 1 Report

This paper shows improvement of CNGM with adalimumab and dapsone therapy, whereas steroids and antibiotics failed.

The introduction is very long and should be shorten. In particular, the paragraph describing the treatment (from line 53), may be postponed in the discussion.

One or more clinical pictures may be useful.

Showing just one of pathology

Author Response

Good morning,

Thank you for your thoughtful comments and edits. In reguards to the three comments provided:

  1. The introduction is very long and should be shorten. In particular, the paragraph describing the treatment (from line 53), may be postponed in the discussion. - Introduction shorted substantially, including the treatment paragraph. Also some content moved into the discussion section.

2. One or more clinical pictures may be useful. -Included as figure 2.

3. Showing just one of pathology- Done

Thank you again!

Reviewer 2 Report

The authors described combination therapy of adalimumab with dapsone against CNGM.

Your case reports are informative for CNGM treatment. I have the following questions.

1.      Do you recommend using antibiotics at first in the case of no evidence of bacteria?

2.      Discuss about suppression of TNF-alpha caused by Dapsone.

Author Response

Good morning, 

Thank you for your thoughtful comments and feedback. In regards to your specific questions: 

  1. Do you recommend using antibiotics at first in the case of no evidence of bacteria? Yes, we do recommend antibiotic therapy regardless of bacterial status. This was a great point that I included in the manuscript- line 120. 
  2. Discuss about suppression of TNF-alpha caused by Dapsone. My understanding is that dapsone works on the anti-neutrophilic pathways with PABA, and does not suppress TNF-alpha- clarification provided in line 149.

Round 2

Reviewer 1 Report

Interesting case report, much improved after revision and reviewer' suggestions.